# Differential Responses of *Medicago truncatula* NLA Homologs to Nutrient Deficiency and Arbuscular Mycorrhizal Symbiosis

**DOI:** 10.3390/plants12244129

**Published:** 2023-12-11

**Authors:** Wei-Yi Lin, Hsin-Ni Yang, Chen-Yun Hsieh, Chen Deng

**Affiliations:** 1Department of Agronomy, National Taiwan University, Taipei 106319, Taiwan; r10621102@ntu.edu.tw (H.-N.Y.); chenyunhsieh1995@gmail.com (C.-Y.H.); 2Department of Horticulture and Landscape Architecture, National Taiwan University, Taipei 106319, Taiwan; jascha1001@gmail.com

**Keywords:** NITROGEN LIMITATION ADAPTATION, *Medicago truncatula*: phosphate transporter, arbuscular mycorrhizal fungi, alternative splicing

## Abstract

NITROGEN LIMITATION ADAPTATION (NLA), a plasma-membrane-associated ubiquitin E3 ligase, plays a negative role in the control of the phosphate transporter family 1 (PHT1) members in Arabidopsis and rice. There are three NLA homologs in the *Medicago truncatula* genome, but it has been unclear whether the function of these homologs is conserved in legumes. Here we investigated the subcellular localization and the responses of *MtNLA*s to external phosphate and nitrate status. Similar to AtNLA1, MtNLA1/MtNLA2 was localized in the plasma membrane and nucleus. *MtNLA3* has three alternative splicing variants, and intriguingly, MtNLA3.1, the dominant variant, was not able to target the plasma membrane, whereas MtNLA3.2 and MtNLA3.3 were capable of associating with the plasma membrane. In contrast with *AtNLA1*, we found that *MtNLA*s were not affected or even upregulated by low-phosphate treatment. We also found that *MtNLA3* was upregulated by arbuscular mycorrhizal (AM) symbiosis, and overexpressing *MtNLA3.1* in *Medicago* roots resulted in a decrease in the transcription levels of *STR*, an essential gene for arbuscule development. Taken together, our results highlight the difference between *MtNLA* homologs and *AtNLA1*. Further characterization will be required to reveal the regulation of these genes and their roles in the responses to external nutrient status and AM symbiosis.

## 1. Introduction

Members of the phosphate transporter family 1 (PHT1) are key players in inorganic phosphate (Pi) acquisition from the rhizosphere and in Pi allocation inside the plants [1,2,3]. Through different levels of the regulation of *PHT1* genes, plants can control Pi uptake efficiency and distribution at tissue level to maintain Pi homeostasis and to support growth and development. At the transcript level, PHOSPHATE STARVATION REGULATOR1 (PHR1), a MYB-type transcription factor, is the key regulator of phosphate starvation responses, binding to P1BS binding domains (GnATATnC) on the promoter regions of *PHT1* genes and inducing their expression under low-Pi conditions [4,5,6]. Before PHT1s target the plasma membrane, PHOSPHATE TRANSPORTER TRAFFIC FACILITATOR 1 (PHF1) in the endomembrane system is required to assist PHT1s exiting the endoplasmic reticulum [7,8]. Under Pi-sufficient conditions, PHT1s are tightly controlled through post-translational regulation both in the endomembrane system and in the plasma membrane, to prevent Pi overaccumulation. PHOSPHATE2 (PHO2), a ubiquitin E2 conjugase, is involved in the ubiquitination of PHT1s in the endomembrane system [9,10]. NITROGEN LIMITATION ADAPTATION (NLA), a RING-type ubiquitin E3 ligase, participates in the ubiquitination of PHT1 in the plasma membrane [11,12]. The loss of function of PHO2 or NLA leads to Pi overaccumulation in shoots [9,11,13,14], demonstrating their importance in controlling PHT1 proteins and maintaining Pi homeostasis.

NLA protein harboring an SPX (SYG1/Pho81/XPR1) domain is present in many plant proteins involved in Pi signaling and transport. The SPX domain can be divided into three conserved subdomains of 30–40 amino acids in length, and these subdomains are separated by sequences with a low similarity [15]. Previous studies in yeast and plants have shown that SPX domains are responsible for physical interaction with the target proteins [11,16,17,18,19]. For example, NLA is able to associate with PHT1s via SPX domain, triggering the vacuolar degradation pathway under Pi-sufficient conditions [11,12,20]. SPX proteins mediate low Pi responses via the prevention of PHRs from activating Pi starvation-responsive genes [17,21]. In recent years, the roles of SPX proteins in the regulation of arbuscular mycorrhizal (AM) symbiosis have been revealed both in plants and AM fungi (AMF). These fungi are the beneficial endosymbionts that can colonize the roots of most land plant species and provide mineral nutrients to host plants in exchange for photosynthetic carbon sources [22]. *Medicago truncatula SPX1* and *SPX3* are induced by low Pi treatment and mycorrhization. Loss of function of these two genes significantly reduces fungal colonization efficiency and also represses arbuscule degeneration-related genes, leading to the enrichment of large arbuscules [23]. In contrast, the functional studies showed that tomato and rice SPX proteins play negative roles in the control of AM symbiosis [24,25], suggesting that further characterization is needed to elucidate the determinant of SPX protein function during AM symbiosis. A SPX-domain containing Pi transporter has been identified in *Rhizophagus irregularis*, an AMF species. Its SPX domain is required for Pi-dependent vacuolar degradation and maintenance of AM symbiosis [26].

Multiple localization of NLA has been reported, localized both in the plasma membrane, nucleoplasm and nuclear speckles [11,20,27]. Deletion of the SPX domain interferes with the interaction with PHT1s and blocks the protein-trafficking to the plasma membrane. Further dissection of the importance of each sub-domain in protein localization found that the loss of any SPX subdomain affects nuclear speckle localization [28]. 

In *M. truncatula*, a model legume, nine PHT1 family members were identified through phylogenetic analysis. It has been shown that MtPT1, MtPT2 and MtPT3 are low-affinity transporters and that MtPT5 is a high affinity transporter [29]. Under low Pi conditions, *MtPT5*, *MtPT6*, *MtPT7* and *MtPT8* are induced in shoots, while *MtPT1*, *MtPT2*, *MtPT5* and *MtPT8* are upregulated in roots. *MtPT2* and *MtPT9* are also induced, but the expression levels are relatively low, compared with other PHT1 members [30]. Different from most MtPTs, *MtPT4* is specifically induced by the association with AMF [31]. The intracellular fungal hyphae penetrate root tissues and form a specialized structure called the arbuscule in the inner cortical cells that provides a platform for nutrient exchange between fungi and the host plants [22]. MtPT4 only localizes on the periarbuscular membrane, the specialized plasma membrane that encircles the arbuscule to take up fungal Pi during symbiosis [32]. Pumplin et al. [33] further demonstrated that polar targeting of MtPT4 is mediated by a transient reorientation of secretion. Surprisingly, MtPT4 and LjPT4, homologs of *Lotus japonicus,* are also present in the root tips of non-mycorrhizal roots and are involved in the regulation of lateral root growth in response to low Pi responses, although the detailed mechanism is unclear [34]. In addition to transcriptional regulation by nutrient status or AM symbiosis, little is currently known about the different regulatory mechanisms controlling the stability and targeting of *Medicago* PHT1s.

There are three NLA paralogs in the *Medicago* genome that belong to the Arabidopsis NLA1 clade [35], hinting that *Medicago* NLAs may also possess similar functions. However, their localization and expression in response to Pi status and AM symbiosis remain unclear. In this study, we aimed to characterize *Medicago NLA*s to reveal their potential function in the regulatory network of Pi homeostasis. Our results show that *MtNLA* paralogs were induced by low Pi, in contrast with Arabidopsis *NLA1*. *MtNLA3* was even upregulated in AM fungal-colonized roots. We also found that *MtNLA3* has three splicing variants and that overexpressing *MtNLA3.1*, the major variant, reduced the transcription levels of *STR*, which encodes a fatty acid exporter on the periarbuscular membrane. In summary, our study highlighted the distinct role of *Medicago* NLAs, which are worthy of further characterization to unravel their function in Pi homeostasis and AM symbiosis.

## 2. Results

### 2.1. Comparison of the Sequence of MtNLA Paralogs and Variants 

Three NLA1 paralogs were present in the *Medicago* genome, Medtr7g108840, Medtr8g058603 and Medtr1g088660, which were named as MtNLA1, MtNLA2 and MtNLA3, respectively. Although these three genes are in the same clade, phylogenetic analysis suggested that MtNLA1 and MtNLA2 are much closer than MtNLA3 [35]. The *MtNLA1* gene contains a 5′ UTR (154 bp) and 3′ UTR (191 bp), and *MtNLA2* does not; however, the coding sequences of these two genes are 100% identical, supporting the idea that these two genes were derived from later duplication events. MtNLA3 also has a 5′ UTR (224 bp) and 3′ UTR (274 bp), and the protein sequence is 82% identical to MtNLA1 and MtNLA2 (Figure 1a and Appendix A). Interestingly, we found three variants of *MtNLA3* in the *Medicago* genome database derived from alternative splicing in the first exon and intron and named as *MtNLA3.1*, *MtNLA3.2* and *MtNLA3.3*. In contrast with *MtNLA3.1* and *MtNLA3.3, MtNLA3.2* has only five exons, and the transcriptional start site and the start codon are located in the first intron and the second exon of *MtNLA3.1*, respectively. The length of the transcript is only 1160 bp and encodes a 254 a.a. protein. This is shorter than *MtNLA3.1*, which is 1449 bp in length and encodes a 316 a.a. protein. *MtNLA3.3* has one more exon, present in the first intron of *MtNLA3.1*, that results in its longer transcript (1712 bp) when compared with *MtNLA3.1*, though the length of the protein is the same (Figure 1b and Appendix A).

The SPX domain in the NLA protein can be divided into three conserved subdomains which are separated by sequences with a low similarity [15]. Through sequence alignment with Arabidopsis and rice NLA1, three conserved subdomains were identified in MtNLA1, MtNLA2 and MtNLA3.1 (Appendix A). Due to the loss of the partial coding sequence, the MtNLA3.2 protein only has full-length subdomain 3 and partial subdomain 2. With regard to the MtNLA3.3 protein, the shift in the start codon leads to the loss of subdomain 1 and the difference in the first three amino acid residues in subdomain 2 (Appendix A). The importance of the SPX domain in protein–protein interaction has been demonstrated [11,17,19,21,36]. It is interesting to see the effects of the partial loss of subdomains in MtNLA3 variants on the feature and function of the proteins. 

### 2.2. Subcellular Localization of MtNLA Paralogs and Variants

It has been shown that Arabidopsis NLA1 and rice NLA1 are predominantly localized in the plasma membrane with occasional signals in nuclear speckles, while the rice NLA2 is likely localized in the cytoplasm [11,20]. To understand the effects of peptide sequence variation on *Medicago* NLA paralogs, we examined the subcellular localization of N- and C-terminal green fluorescent protein (GFP)-tagged fusion proteins in the leaf epidermal cells of *Nicotiana benthamiana*. Due to the 100% identity of the MtNLA1 and MtNLA2 protein sequence (Appendix A), the coding sequence was amplified to generate the GFP-fused MtNLA1/MtNLA2 for the following study. Similar to the previous observation [11], GFP-MtNLA1/MtNLA2 was predominantly localized in the plasma membrane while additional nuclear signals were observed when GFP was tagged at the C terminus of MtNLA1/MtNLA2 (Figure 2 and Appendix A). Surprisingly, MtNLA3.1, which harbors three conserved SPX subdomains, was localized in the nucleus when GFP was tagged at the N terminus, whereas fluorescence signals in the cytoplasm and nucleus were observed when tagging at the C terminus (Figure 2 and Appendix A). Although MtNLA3.2 only has a partial subdomain 2 and a conserved subdomain 3, N-terminal GFP-tagged proteins were able to be localized both in the plasma membrane and nucleus while lesser nuclear signals were observed for the C-terminal GFP-tagged proteins. For MtNLA3.3, which harbors two conserved SPX subdomains, we also observed fluorescence signals of N-terminal fusion proteins both in the plasma membrane and nucleus, while C-terminal fusion proteins were found in the punctate structures associated with the plasma membrane (Figure 2 and Appendix A). In summary, the subcellular localization of MtNLA1/MtNLA2 is the same as AtNLA1 and OsNLA1 but the localization of MtNLA3 variants is different: MtNLA3.2 and MtNLA3.3 are predominantly localized in the plasma membrane while MtNLA3.1 is in the nucleus, suggesting that the conservation of subdomain 3 in SPX and the RING domain at the C terminus might be sufficient to determine the plasma membrane and nuclear localization of NLA. Further characterization will be required to understand the influence of sequence variation on subcellular localization.

### 2.3. The Responses of MtNLA Paralogs to Nutrient Deficiency and Arbuscular Mycorrhizal Symbiosis

Arabidopsis *NLA1* is the target of microRNA827, a Pi-starvation-induced small RNA. The post-transcriptional cleavage results in the reduction in *AtNLA1* transcript levels, both in shoots and roots, under Pi-deficient conditions [37]. In contrast, the transcript level is slightly increased under low-nitrate conditions [14]. Different from *AtNLA1*, rice *NLA1* does not have microRNA827 target sites, but the down- and up-regulation of the *OsNLA1* gene, along with the decrease in Pi and nitrate concentration, was still observed in shoots and roots, respectively [38,39]. *Medicago NLA* homologs are not the target of microRNA827 either [35], and their responses to Pi and nitrate deficiency are still unclear. To elucidate the transcriptional regulation of *MtNLA* paralogs in response to nutrient deficiency, *Medicago* seedlings were transferred to hydroponic solution containing different levels of Pi or nitrate. The Pi concentration in leaves under low-Pi conditions were significantly lower than that under control and low nitrate conditions (Figure 3a). MtNRT2.1, a major high-affinity nitrate transporter, is the major contributor to nitrate uptake in the roots under nitrogen-limited conditions [40]. The transcript levels of *MtNRT2;1* were increased under low-nitrate treatments (Figure 3b). Both phenotypes supported the idea that our treatments were effective. Due to the high sequence similarity between *MtNLA1* and *MtNLA2*, we designed the same primer set to amplify these two genes simultaneously using real-time quantitative PCR (qPCR) (Figure 1a). The expression of *MtNLA1/MtNLA2* in shoots was upregulated under low-Pi conditions but the transcript levels under low-nitrate or low-nitrate/low-Pi conditions were similar to those under control conditions. In the roots, the expression levels of *MtNLA1/MtNLA2* were also slightly but not significantly increased by low-Pi treatments (Figure 3c). To analyze the overall transcription levels of *MtNLA3* variants, we designed a primer set against the 3′ end of the *MtNLA3* coding region to amplify all of the variants simultaneously (Figure 1b). Different from *MtNLA1/MtNLA2*, the transcript levels of *MtNLA3* in the shoots were only induced by low-Pi/low-nitrate treatments, while in the roots, the expression of *MtNLA3* was upregulated under low-Pi and low-Pi/low-nitrate conditions, when compared with control conditions (Figure 3d). To further elucidate the difference in *MtNLA3* variants in response to nutrient treatments, we performed real-time qPCR using three different primer sets to specifically amplify three variants (Figure 1b). The expression pattern of *MtNLA3.1* in roots coincided with that of *MtNLA3*, and in shoots, *MtNLA3.1* was significantly induced by low-nitrate and low-nitrate/low-Pi treatments. In contrast, the transcription levels of *MtNLA3.3* were not affected by any nutrient treatments in shoots but upregulated under low-nitrate conditions (Figure 3e,f). The expression of *MtNLA3.2* was too low to be detected in this condition. PHR1, an MYB-type transcription factor, is one of the well-characterized regulators of Pi starvation-responsive genes via binding to the PHR1 binding site (P1BS) on the promoter regions of target genes [41,42]. The analysis of *cis*-acting elements showed that there are three P1BS motifs in the *MtNLA3* gene, but none were identified in the *MtNLA1* and *MtNLA2* genes (Appendix A). *OsNLA1* has one P1BS in the promoter region, but its expression does not depend on OsPHR2 [20]. Further study is required to see whether PHR1 mediates the upregulation of *MtNLA3* in roots. 

Based on the expression pattern shown in the *Medicago truncatula* gene expression atlas web server [43], the transcript levels of three *MtNLA* paralogs were not affected by arbuscular mycorrhizal symbiosis, but the mRNA level of *MtNLA3* was higher in arbuscule-containing cortical cells than in adjacent cortical cells and non-colonized cortical cells (Appendix A), implying that *MtNLA3* might play a role in AM symbiosis. To confirm the responses of *MtNLA* paralogs to AMF colonization and the effects of the AMF species on gene expression, we examined their transcript levels in roots colonized by two different AMF species, *Claroideoglomus etunicatum* and *Rhizophagus irregularis*. The colonization efficiency of these two AMF species have been shown previously [44]. Here we further examined the expression of marker gene, *MtBCP1* [45], and found similar induction levels of this gene by two different AMF species (Figure 4a), indicating that there is no difference of colonization efficiency by these two fungal species. In agreement with the expression pattern shown in the *Medicago* gene atlas, the transcription levels of *MtNLA1/MtNLA2* were not affected by symbiosis. Intriguingly, *MtNLA3* was induced by *R. irregularis* colonization but not by *C. etunicatum*. (Figure 4b) We further dissected the expression levels of *MtNLA3* variants during symbiosis. Although the transcript levels of *MtNLA3.2* were relatively lower than those of *MtNLA3.1* and *MtNLA3.3*, all three variants were induced by *R. irregularis* colonization, but not by *C. etunicatum* (Figure 4c). Our results suggest that *MtNLA3* might function in AM fungal-colonized roots, though the response of *MtNLA3* to symbiosis varies with the AMF species. 

### 2.4. Interaction between MtNLAs and MtPTs

NLA is known as a post-translational regulator of PHT1s to prevent Pi overaccumulation. It has been shown that the SPX domain of AtNLA1 can interact with AtPHT1.1 and AtPHT1.4 on the plasma membrane [11]. According to the phylogenetic analysis, MtPT1, MtPT2 and MtPT3 are the closest homologs to AtPHT1.1 [30]. We performed a split-ubiquitin yeast two-hybrid assay to examine the interaction between MtNLA paralogs and MtPT1. MtNLA3.1, which cannot target the plasma membrane in *N. benthamiana* leaves, was unable to interact with MtPT1, using either the full-length or solely the SPX domain as the prey (Figure 5a). Although both MtNLA3.2 and MtNLA3.3 were able to localize in the plasma membrane, only MtNLA3.3 could interact with MtPT1. Subcellular localization of MtNLA1/MtNLA2 was similar to AtNLA1 and OsNLA1 (Figure 2), and we did observe the interaction between MtPT1 and MtNLA1 in yeast (Figure 5b). Unexpectedly, the protein–protein interaction was not able to be validated via bimolecular fluorescence complementation test in tobacco leaves. Further validation will be required to confirm the interaction in planta. 

MtPT4 is known as an AM symbiotic-induced PHT1, responsible for fungal Pi uptake in arbuscule-containing cells [31,32]. All of the *MtNLA3* variants were induced by *R. irregularis* colonization (Figure 4c) and relatively high expression in arbuscule-containing cortical cells (Appendix A). We speculated that MtNLA3 might function as a regulator of MtPT4. To address this, MtPT4 was used as bait to test its interaction with MtNLA paralogs. Surprisingly, none of the MtNLA3 variants or MtNLA1 were able to interact with MtPT4 in the yeast two-hybrid system (Figure 5), suggesting that MtNLAs might not be involved in the control of PHT1 in arbuscule-containing cells, even though it was positively regulated by symbiosis. 

### 2.5. The Phenotype of MtNLA3-Overexpressing Roots in Response to Arbuscular Mycorrhizal Symbiosis 

In order to characterize the function of MtNLA3 variants during AM symbiosis, we generated composite roots to overexpress the coding sequences of *MtNLA3.1*, *MtNLA3.2* and *MtNLA3.3* in *Medicago* using *Agrobacterium rhizogenes*-mediated root transformation. After a six week inoculation by *R. irregularis*, composite roots were harvested to examine the expression of the *R. irregularis α-Tublin* (*RiTub*) gene and several marker genes activated in the symbiotic signaling pathway and arbuscule development, including *MtIPD3* [46], *MtDMI3* [47], *MtVapyrin* (*MtVPY*) [48,49], *MtRAM1* [50,51], *MtSTR* [52], *MtPT4* and *MtBCP1* [45] to elucidate the effects of overexpressing target genes on AMF colonization. In *MtNLA3.1*-overexpressing roots, the transcript levels increased more than 10 fold, compared with the empty control (Ev), but the frequency of colonization was not affected (Figure 6a,b). Coincidently, fungal staining showed the similar colonization phenotype in Ev and overexpressing roots (Figure 6c). In terms of symbiotic marker genes, the transcript levels of *MtSTR* were significantly downregulated, and *MtRAM1*, *MtPT4* and *RiTub* were also slightly reduced by the increase in *MtNLA3.1* mRNA; however, the expression levels of *MtIPD3*, *MtVPY* and *MtBCP1* were not affected (Figure 6d–g). On the other hand, *MtNLA3.2*- or *MtNLA3.3*-overexpressing roots increased the expression levels of *MtNLA3* by 50 to 100 fold, compared with the control (Figure 7a). However, the phenotype of fungal colonization and the expression levels of most symbiotic marker genes we examined were not affected by the transgenes and we even observed the significant upregulation of *MtBCP1* in *MtNLA3.2*-overexpressing roots (Figure 7). Although *MtNLA3* variants were also upregulated by AMF colonization, further characterization is required to elucidate their function in symbiosis.

## 3. Discussion

Although the roles of NLA1 in the post-translational control of PHT1s have been demonstrated in Arabidopsis and rice, the regulation of the *NLA1* gene was different in these two model plant species [11,20]. *AtNLA1* is under micorRNA827-mediated post-transcriptional regulation [11,14], while *OsNLA1* is regulated by its promoter and upstream open reading frame [39]. However, there have been no reports regarding the regulation and function of *NLA1* paralogs in other species. *M. truncatula*, a model legume, has three NLA1 paralogs due to the duplication events during plant evolution [35]. Moreover, MtNLA3 has three alternative splicing variants. It has been shown that the sequence deletion or insertion in alternative splicing variants may lead to different localization and functions [53]. In this study, we investigated the expression of *NLA1* paralogs in *Medicago* with regard to external nitrate and Pi levels. Similar to the responses of *AtNLA1* to low-nitrate treatment, the expression of *MtNLA3* was slightly increased in shoots but not in roots. Intriguingly, the transcription levels of *MtNLA1/MtNLA2* in shoots and *MtNLA3* in roots were also upregulated by low-Pi treatments and the transcription levels of *MtNLA3* were even higher under low-Pi/low-nitrate conditions (Figure 3), suggesting that the regulation of *MtNLA* paralogs by external nutrient supply was different from Arabidopsis and rice. In fact, Yue et al. [20] determined the transcript levels of *OsNLA1* were not significantly affected by low-Pi treatment, which is in contrast to the studies by Yang et al. [38] and Yang et al. [39], suggesting that there might be some unknown factors involved in the control of *NLA* expression. Expression quantitative trait locus (eQTL), a region of DNA which contributes to the variation in genetic regulation and accounts for the diversity of phenotype, is important for the control of crop production and breeding. For example, the upstream open reading frame on *OsNLA1*, which is crucial for Pi-responsive regulation, was only identified in rice cultivars but not in wild rice species, implying that this fragment was adopted after rice domestication [39]. Through eQTL mapping, the variation in a cruciferin gene expression in a potato population was identified that is highly correlated to tuber starch content. The difference in gene expression between cultivars is attributed to the variation in *cis*-regulatory elements and the binding efficiency of trans-regulatory factors [54]. Carrasco-Valenzuela et al. [55] used a similar strategy and found that genetic variation alters auxin levels, which affects the fruit-softening rate in peaches. *MtNLA3* gene harbors three P1BS motifs and a 120 bp upstream open reading frame that expands from the promoter to the coding region (Figure 1 and Appendix A). Further studies are required to identify other unknown regulatory factors and to elucidate the influence of these regulatory elements on gene expression.

Although the responses of *MtNLA*s to low-Pi treatment were different from *AtNLA1*, the subcellular localization of MtNLA1/MtNLA2 and the two MtNLA3 variants, MtNLA3.2 and MtNLA3.3, were similar to AtNLA1 and OsNLA1. A bipartite nuclear localization signal (NLS) at the N terminus was conserved in Arabidopsis, rice and *Medicago* NLA1 paralogs (Appendix A). Substitution of the positively charged residues in the NLS or deletion of any SPX subdomains in AtNLA1 disrupted the protein functions and nucleoli and nuclear speckle localization but did not affect protein targeting to the plasma membrane [28]. If the whole SPX domain is deleted, the truncated proteins are unable to associate with the plasma membrane [11]. These results indicate that the N-terminus of NLA1 determines the protein localization in subnuclear doamin and the plasma membrane. There is no transmembrane domain in NLA1 proteins. It is possible that the association of NLA1 homologs with the plasma membrane is due to the interaction with the membrane protein through the SPX domain. MtNLA3.1 obtained three subdomains and an NLS, which are highly conserved in NLA homologs (Appendix A), but it was localized in nucleus and/or cytoplasm when overexpressed in the leaf epidermal cells of *N. benthamiana* (Figure 2). However, MtNLA3.2 and MtNLA3.3, which have only one and two SPX subdomains, respectively, were able to localize in the plasma membrane and nucleus (Figure 2). Whether the lack of proper membrane protein in *N. benthamiana* cells or the presence of first SPX subdomain affects the protein localization requires further validation in *Medicago* roots. In addition, the results of the split-ubiquitin Y2H assay showed that only MtNLA3.3 and MtNLA1/MtNLA2 can interact with MtPT1, a Pi-starvation-induced PHT1 family member (Figure 5). Combined with the findings of Hannam et al. [28] and our observation of the subcellular localization of MtNLA3 variants, we suggested that the presence of the second and third SPX subdomains was enough for plasma membrane targeting and to interact with PHT1. 

It is worth noting that *MtNLA3* variants were significantly induced in *R. irregularis-* colonized roots but not in *C. etunicatum*-colonized roots while the transcription levels of *MtNLA1/MtNLA2* were not affected by AMF colonization (Figure 4). Our results imply that MtNLA3 might play a role in AM symbiosis and its importance might be varied by fungal species. The differential expression of symbiotic-responsive genes when colonized by different AMF species has been shown in *Medicago*, cassava and sorghum [56,57,58], suggesting that the difference in fungal species has a great impact on the interaction between host plant and fungi in terms of the molecular aspect. In addition, we observed a decrease in transcription levels of the fungal tubulin genes, *MtSTR* and *MtPT4,* which are involved in arbuscule development in *MtNLA3.1-*overexpressing roots, but not in *MtNLA3.2-* or *MtNLA3.3-*overexpressing roots (Figure 6 and Figure 7), hinting that MtNLA3.1 might play a negative role in the regulation of arbuscule development process. However, the colonization phenotype in overexpressing roots were as normal as they were in the EV control. We speculate that the proteins that remained in the overexpressing roots were sufficient to support arbuscule development. Further investigation will be performed to identify its interacting partners and elucidate the role of MtNLA3 during AM symbiosis.

Ubiquitination is one of the post-translation regulatory mechanisms that affects the stability or trafficking of target proteins [59]. Many AM symbiotic-induced genes have been identified when colonized in roots, but very few studies have revealed the regulation of symbiotic responses at the protein level. PUB1, a ubiquitin E3 ligase, is the only known post-translational regulator to be involved in the modulation of initiation of endosymbiosis through interaction with DMI2, a receptor in common symbiotic signaling pathway [60]. In this study, we showed that MtNLA3 is another ubiquitin E3 ligase which might be involved in the regulation of genes related to arbuscule development. Further studies will be required to identify the target proteins and to elucidate their role in the regulatory network.

In conclusion, our findings show the differential responses of *Medicago NLA* paralogs to low Pi, compared with Arabidopsis and rice. Moreover, the induction of *MtNLA3* by AM symbiosis and the downregulation of genes involved in arbuscule development in *MtNLA3.1* overexpressing roots implied the potential role of MtNLA3 in symbiosis. However, the subcellular localization of MtNLA3.1 is in the nucleus, in contrast with the dual localization of AtNLA1 and other MtNLA paralogs. Further characterization is required to reveal the regulation and function of NLAs in legume species.

## 4. Materials and Methods

### 4.1. Plant Growth Conditions

Plants used in this study were grown in a growth chamber with 80.5 μmole m^−2^ s^−1^ light intensity, a 16 h light (25 °C) and 8 h dark (22 °C) cycle and 60–70% relative humidity. *Medicago truncatula* ssp. *truncatula* ecotype Jemalong (A17) seeds were surface sterilized by 10% bleach and kept at 4 °C for two days before germination. For hydroponic growth, seven-day old seedlings were transplanted into the hydroponic culture system filled with a modified half-strength Hoagland’s solution. For low-Pi and low-nitrate treatment, the media contained only 20 μM potassium phosphate and 0.75 mM NO_3_^−^, respectively, one-tenth the level present in nutrient-sufficient media. The media were refreshed every week. Plants were harvested at 3 weeks after transplantation and stored at −80 °C for Pi concentration measurement and RNA extraction. For AM symbiosis, seedlings were transplanted to cones filled with sterilized river sand containing 1 g of *C. etunicatum* or *R. irregularis* inoculants (approximately 100 spores). Seedlings were fertilized twice a week with low-Pi Hoagland’s solution containing 20 μM Pi. Plants were harvested at 6 weeks after inoculation for further research. 

*Nicotiana benthamiana* were used for *Agrobacterium tumefaciens*-mediated infiltration. Plants were grown in pots filled with a mixture of peat and vermiculite in a 9:1 ratio and fertilized with full nutrient Hoagland’s solutions once a week. Leaves were ready for infiltration four weeks after germination. 

### 4.2. Plasmid Construction

The coding sequence of the genes of interest were amplified using PCR and cloned to pDONR221 via a Gateway BP reaction (Thermo Fisher Scientific, Waltham, MA, USA). Primer sequences used in this study are listed in Appendix A. For generating N- and C-terminal GFP-tagged proteins driven by CaMV 35S promoter, genes were cloned to pK7WGF2 and pK7GWF2, respectively [61], through an LR recombination reaction (Thermo Fisher Scientific). 

To generate gene-overexpressing constructs, cloning vectors that obtained the genes of interest were recombined with a pKm43GW-RedRoot destination vector which contains *DsRed* as a marker gene [62]. The CaMV 35S promoter was used to drive *DsRed* and the genes of interest.

To generate constructs for the split-ubiquitin membrane yeast two-hybrid assay, the coding sequence of *MtNLA1*, *MtNLA3* variants and SPX domain of *MtNLA3.1* (1–210 a.a.) were cloned to the *BamH*I and *EcoR*I sites of the prey vectors, pDL-Nx and pDL-xN. The coding sequence of *MtPT1* and *MtPT4* were cloned to the *Xba*I and *Stu*I sites, respectively, of the bait vector, pAMBV4 (Dualsystems Biotech, Schlieren, Switzerland). 

### 4.3. Agrobacterium Tumefaciens-Mediated Infiltration

*Agrobacterium tumefaciens*-mediated infiltration was conducted as described by Liu et al. [36] with minor modification. The overnight culture of the *A. tumefaciens* EHA105 strain containing a construct of the gene of interest was prepared in Luria–Bertani medium. Cells were harvested by centrifugation and resuspended in the infiltration medium (10 mM MgCl_2_, 10 mM MES buffer and 100 μM acetosyringone) to an OD_600_ = 1. The cell suspension was kept in the dark for 2–3 h and then, was used to infiltrate the leaves of *N. benthamiana* using needleless syringes. Leaves were harvested at 3 or 4 days after infiltration for research.

### 4.4. Agrobacterium Rhizogenes-Mediated Root Transformation

*Agrobacterium rhizogenes*-mediated root transformation was performed as described previously [63]. Briefly, *Medicago* seeds were surface sterilized and kept at 4 °C overnight. Then, the seeds were germinated in the dark at 30 °C for 15 h. The radicles were cut and were coated with *A. rhizogenes* strain Arqua1 containing a construct of the gene of interest. After 3 weeks of cocultivation, seedlings with DsRed expression in composite roots were transplanted to cones filled with sterilized river sand and 1 g of *R. irregularis* inoculants. Plants were grown for an additional six weeks before harvesting.

### 4.5. Split-Ubiquitin Membrane Yeast Two-Hybrid Assay

Split-ubiquitin membrane yeast two-hybrid assay was performed according to the manufacturer’s instruction (Dualsystems Biotech). Briefly, a bait and a prey vector were co-transformed to yeast strain DSY1 cells (*ura3 trp-1 leu2 his3 GAL1::HIS3 GAL::lacZ GAL2::Ade3*) and the cells were grown on a synthetic medium lacking leucine and tryptophan. Specificity of the protein–protein interaction was confirmed by growth on a synthetic medium lacking leucine, tryptophan and histidine.

### 4.6. WGA Staining and Analysis of Colonization Efficiency

Fungal structures in *Medicago* roots were stained with WGA-Alexa fluor 488 (Thermo Fisher Scientific) as described previously [51]. The root fragments were selected randomly and examined microscopically using an Olympus SZX16 stereomicroscope (Olympus, Tokyo, Japan). The colonization efficiency was evaluated by the grid method [64].

### 4.7. Confocal Microscopy

Confocal microscope images were taken using a Leica TCS SP5 II (Leica Microsystems, Germany) with the objectives HCX PL FLUOTAR 10×/0.30 DRY and HCX PL APO CS 20×/0.7 dry. Excitation and emission wavelengths for GFP were 475 nm and 500–520 nm, respectively. 

### 4.8. RNA Isolation and Gene Expression Analysis

Total RNA was isolated using TRIzol reagent (Thermo Fisher Scientific) according to the manufacturer’s instructions, followed by TURBO DNase I (Thermo Fisher Scientific) treatment to remove genomic DNA. Complementary DNA was synthesized from 500 ng RNA using Moloney murine leukemia virus reverse transcriptase (Thermo Fisher Scientific). Gene expression analysis was performed using an iQ^TM^ SYBR^®^ Green Supermix (Bio-Rad, Hercules, CA, USA) on a CFX connected real-time PCR detection system (Bio-Rad). The relative expression levels were normalized to the expression of a reference gene, *MtEF1*-α. Primer sequences are listed in Appendix A.

### 4.9. Pi Concentration Measurement

Pi concentration measurement was performed as described by Chiou et al. [65]. 

### 4.10. Statistical Analysis

At least three biological replicates were sampled for analysis. The difference between the control and treatments was evaluated using ANOVA with Tukey’s HSD test for multiple comparison analysis or Student’s *t* test for paired data analysis. 

## Figures and Tables

**Figure 1 plants-12-04129-f001:**
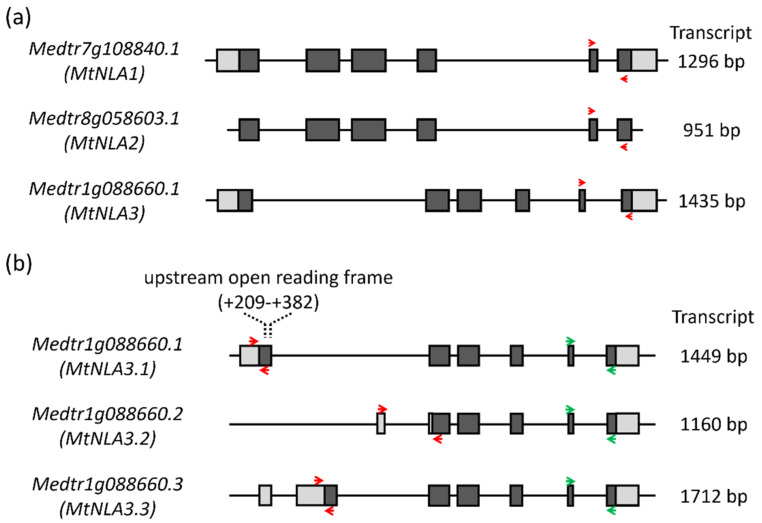
The gene structure of *MtNLA* paralogs (**a**) and *MtNLA3* variants (**b**). Light and dark gray boxes indicate untranslated and coding regions, respectively. Red arrows in (**a**) indicate the location of specific primers used for detecting the expression of *MtNLA* genes. Red arrows in (**b**) indicate specific primers used for detecting the expression of *MtNLA3* variants. Green arrows indicate the common primers used for detecting the expression of all *MtNLA3* variants simultaneously.

**Figure 2 plants-12-04129-f002:**
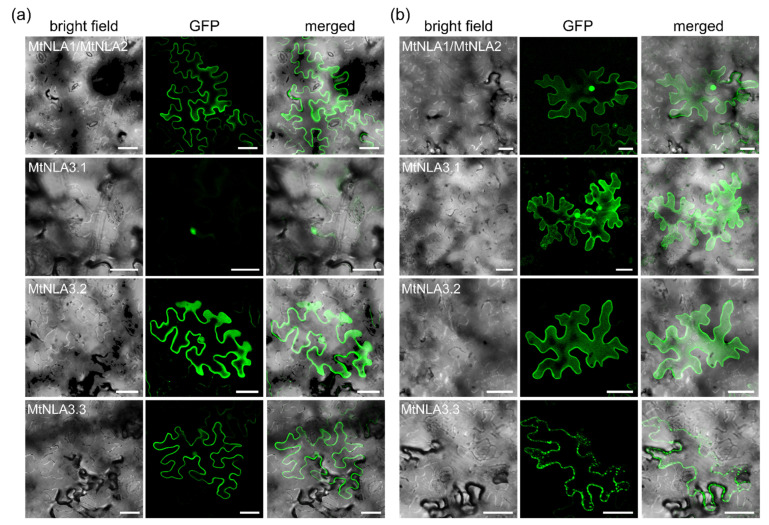
The subcellular localization of MtNLAs. GFP was tagged at either the N-terminus (**a**) or C-terminus (**b**) of NLA proteins variants to reflect the localization of the target. Merged images are the overlay of GPF and bright field. Bar = 50 μm.

**Figure 3 plants-12-04129-f003:**
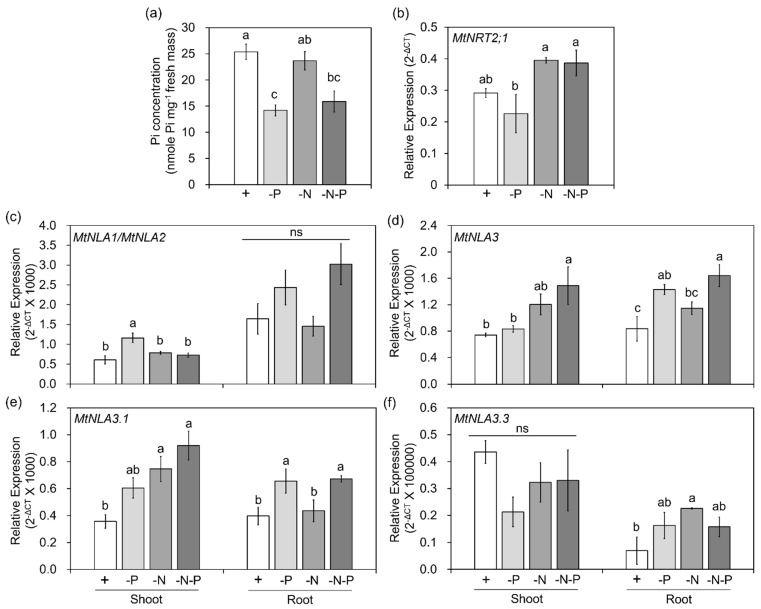
The responses of *MtNLA* genes to low-Pi (-P; 20 μM Pi) or low-nitrate treatments (-N; 0.75 mM ${NO}_{3}^{-}$. (**a**) The shoot Pi concentration. (**b**) Relative expression of *MtNRT2.1.* (**c**–**f**) Relative expression of *MtNLA1/MtNLA2* (**c**), *MtNLA3* (**d**), and *MtNLA3* variants (**e**,**f**). MtEF1 was used as a reference gene for normalization. N = 3. Values are mean ± SE. Data were analyzed by ANOVA with Tukey’s HSD test (*p* < 0.05). Different letters indicate significant difference. ns indicates that the difference was not significant.

**Figure 4 plants-12-04129-f004:**
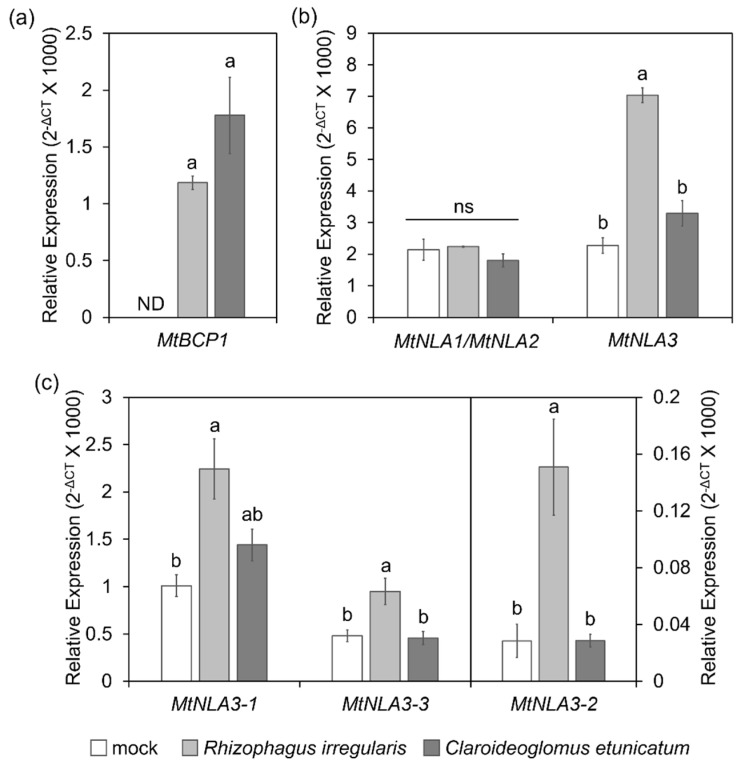
Relative expression of a symbiotic marker gene, *MtBCP1* (**a**), *MtNLA1/MtNLA2*, *MtNLA3* and *MtNLA3* variants (**b**,**c**). N = 4. *MtEF1* was used as a reference gene for normalization. Values are mean ± SE. ND indicates not detectable. Data were analyzed by ANOVA with Tukey’s HSD test (*p* < 0.05). Different letters indicate significant difference. ns indicates a difference that was not significant.

**Figure 5 plants-12-04129-f005:**
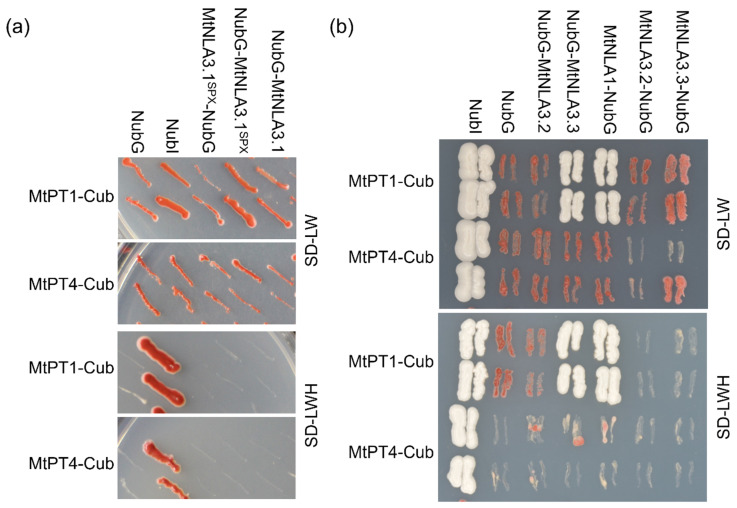
The interaction test between MtNLAs and MtPT1 or MtPT4 using the split-ubiquitin yeast two-hybrid system. (**a**) The interaction test of full-length or the SPX domain of MtNLA3.1 either with MtPT1 or MtPT4. (**b**) The interaction test of MtNLA3 variants or MtNLA1/MtNLA2 with MtPTs. The interaction was evaluated by cell growth on drop-out media lacking leucine (L), tryptophan (W) and histidine (H). The interactions with NubG and NubI were used as the negative and positive control, respectively.

**Figure 6 plants-12-04129-f006:**
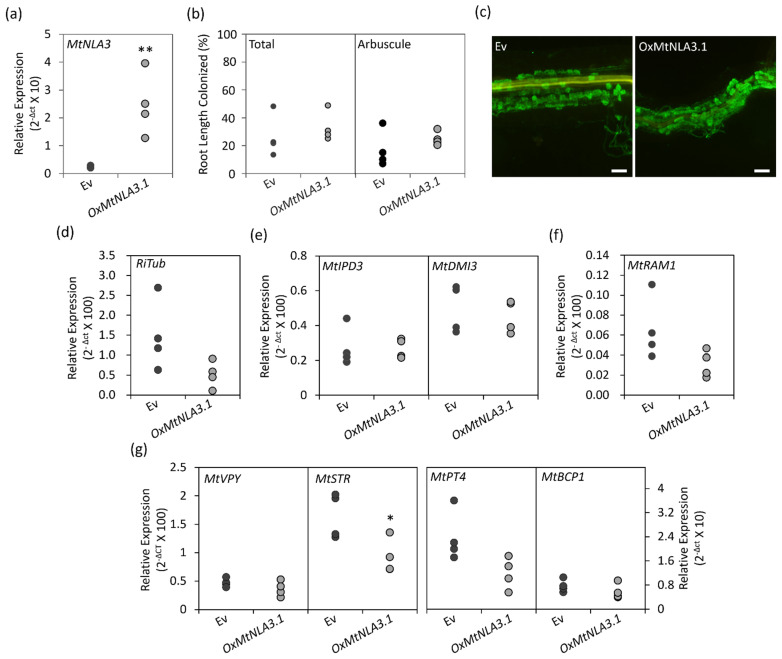
The symbiotic phenotypes in *MtNLA3.1*-overexpressing roots. (**a**) Relative expression of *MtNLA3*. (**b**) The frequency of fungal colonized roots and the percentage of arbuscule-containing root fragments in total root fragments observed. (**c**) The fungal staining of colonized root fragments. Bar = 100 μm. (**d**–**g**) Relative expression of *RiTub* (**d**) and marker genes of symbiotic signaling (**e**) and arbuscule development (**f**,**g**) in empty vector control (Ev) and *MtNLA3.1*-overexpressing roots (*OxMtNLA3.1*). *MtEF1* was used as a reference gene for normalization. N = 4. Significant difference was evaluated using Student’s *t* test. *, *p* < 0.05, **, *p* < 0.01.

**Figure 7 plants-12-04129-f007:**
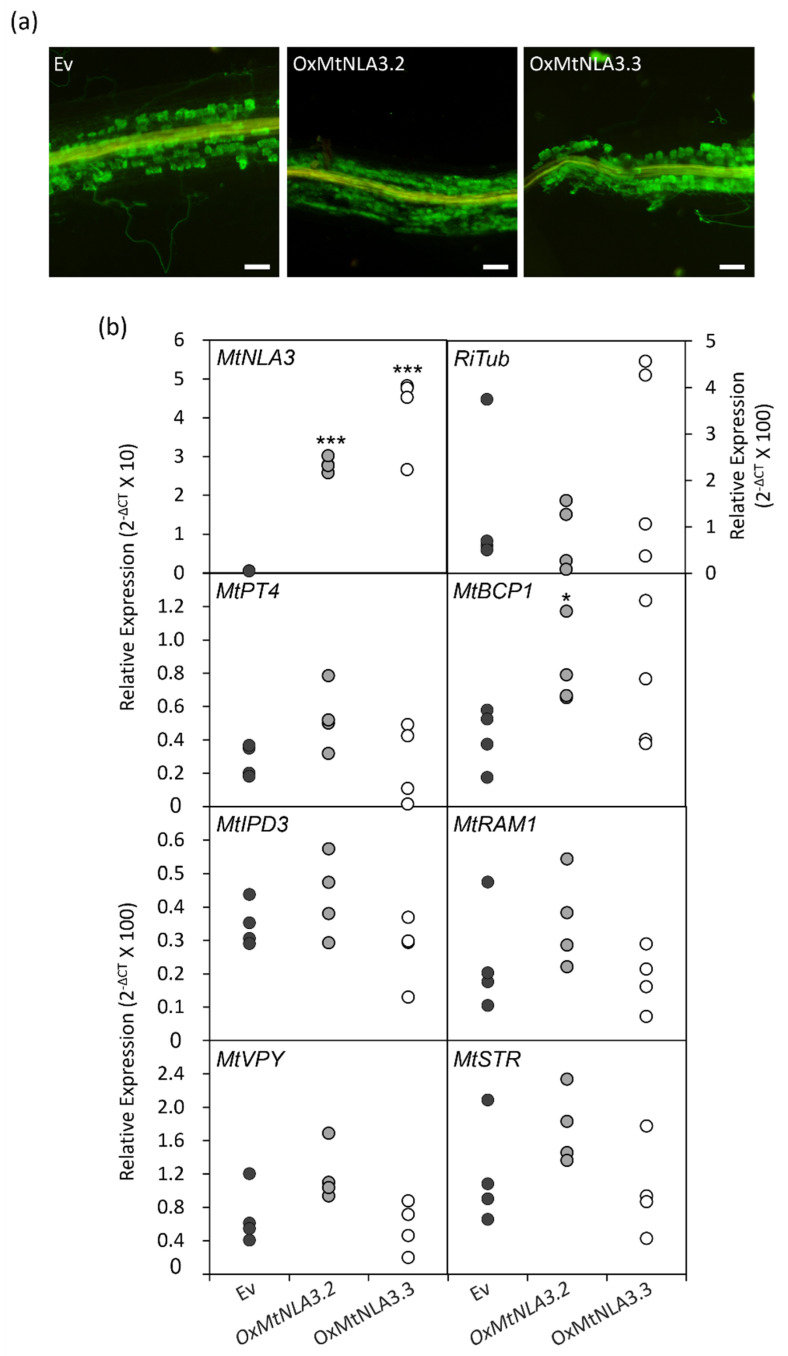
The fungal staining (**a**) and relative expression of *MtNLA3, RiTub* and symbiotic marker genes (**b**) in empty vector control (Ev), *MtNLA3.2-* and *MtNLA3.3*-overexpressing roots (*oxMtNLA3.2* and *OxMtNLA3.3,* respectively). Bar = 100 μm. *MtEF1* was used as a reference gene for normalization. N = 4. Significant difference was evaluated using Student’s *t* test. *, *p* < 0.05, ***, *p* < 0.001.

## Data Availability

All data supporting the findings of this study are available within the paper and its Appendix A.

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
