# Peer review of "Differential Responses of Medicago truncatula NLA Homologs to Nutrient Deficiency and Arbuscular Mycorrhizal Symbiosis"

_plants, 2023, doi:10.3390/plants12244129_

Round 1

Reviewer 1 Report

Comments and Suggestions for Authors

Comments to the authors:

Wang et al. submitted the draft titled by “Differential responses of Medicago truncatula NLA homologs to nutrient deficiency and arbuscular mycorrhizal symbiosis”. This is an interesting study. The results are completely presented in this MS. However, some references need to be cited in the introduction, and the pictures in results need to be further recombined before publication. The detail comments are shown as below:

Detail comments:

1.     The first passage in the introduction section needs to be improved: authors should also introduce the PHT1 transporters’ roles in AM symbioses, therefore, some important research articles on AM-inducible PT4 transporters are lacking. For examples, (1) Pumplin et al., 2012. Polar localization of a symbiosis-specific phosphate transporter is mediated by a transient reorientation of secretion. PNAS. 109(11): E665-672. (2) Xie et al., 2013. Functional analysis of the novel mycorrhiza-specific phosphate transporter AsPT1 and PHT1 family from Astragalus sinicus during the arbuscular mycorrhizal symbiosis. New Phytol. 198(3): 836-852. (3) Volpe et al., 2016. The phosphate transporters LjPT4 and MtPT4 mediate early root responses to phosphate status in non mycorrhizal roots. Plant, Cell & Environment 39: 660-671.

2.     L49-58: in the introduction section, the advances of SPX proteins in AM symbioses are lacking, for example, (1) Wang et al., 2021. Medicago SPX1 and SPX3 regulate phosphate homeostasis, mycorrhizal colonization, and arbuscule degradation. Plant Cell 33: 3470-3486. (2) Shi et al., 2021. A phosphate starvation response-centered network regulates mycorrhizal symbiosis. Cell 184: 5527-5540. (3) Xie et al.,2022. A SPX domain-containing phosphate transporter from Rhizophagus irregularis handles phosphate homeostasis at symbiotic interface of arbuscular mycorrhizas. New Phytol. 234, 650-671.

3.     Fig.2,3,5,6: in the legends, the reference gene need to be shown.

4.     Fig. 5a: only the frequency of AMF colonization?? Why did not the authors provide the mycorrhizal intensity (M%), and arbuscule abundance (A%)?

5.     Fig.S1: The gene structure of MtNLAs is an important result, which should be placed into the Fig.1 in the main text, whereas Fig.2 is the subcellular localization of MtNLAs, and so on.

6.     Fig.S6: The AM phenotype of MtNLA3-overexpression should be removed into Fig.5, and further recombine this figure.

Comments on the Quality of English Language

Minor editing of English language required

Author Response

Dear Reviewer,

We are grateful for the comments and suggestions to improve our manuscript. Here are our point-by-point responses to your comments.

  1. The first passage in the introduction section needs to be improved: authors should also introduce the PHT1 transporters’ roles in AM symbioses, therefore, some important research articles on AM-inducible PT4 transporters are lacking. For examples, (1) Pumplin et al., 2012. Polar localization of a symbiosis-specific phosphate transporter is mediated by a transient reorientation of secretion. PNAS. 109(11): E665-672. (2) Xie et al., 2013. Functional analysis of the novel mycorrhiza-specific phosphate transporter AsPT1 and PHT1 family from Astragalus sinicusduring the arbuscular mycorrhizal symbiosis. New Phytol. 198(3): 836-852. (3) Volpe et al., 2016. The phosphate transporters LjPT4 and MtPT4 mediate early root responses to phosphate status in non mycorrhizal roots. Plant, Cell & Environment 39: 660-671.
    Our Response: Thank you for your suggestion. We have added the information as suggested (line 86-90).
  2. L49-58: in the introduction section, the advances of SPX proteins in AM symbioses are lacking, for example, (1) Wang et al., 2021. Medicago SPX1 and SPX3 regulate phosphate homeostasis, mycorrhizal colonization, and arbuscule degradation. Plant Cell 33: 3470-3486. (2) Shi et al., 2021. A phosphate starvation response-centered network regulates mycorrhizal symbiosis. Cell 184: 5527-5540. (3) Xie et al.,2022. A SPX domain-containing phosphate transporter from Rhizophagus irregularishandles phosphate homeostasis at symbiotic interface of arbuscular mycorrhizas. New Phytol. 234, 650-671.
    Our Response: Thank you for your suggestion. We have added the information as suggested (line 54-69).
  3. Fig.2,3,5,6: in the legends, the reference gene need to be shown.
    Our Response: We have added the reference gene used for normalization in the legends.
  4. Fig. 5a: only the frequency of AMF colonization?? Why did not the authors provide the mycorrhizal intensity (M%), and arbuscule abundance (A%)?
    Our Response: Thank you for your comments. Comparing to M% and A% which need to evaluate the percentage of mycorrhization and arbuscule abundance by personal jugement, we think that F% is an objective indicator of colonization efficiency. Combined with microscopy and molecular data, we believed that it is enough to show the effects of treatments or transgenes on mycorrhization.

Reference: Trouvelot A, Kough JL & Gianinazzi-Pearson V (1986) Mesure du taux de mycorhization VA d’un système radiculaire. Recherche de méthodes d’estimation ayant une signification fonctionnelle. In : Physiological and Genetical Aspects of Mycorrhizae, V. Gianinazzi-Pearson and S. Gianinazzi (eds.). INRA Press, Paris, pp. 217-221.

  1. Fig.S1: The gene structure of MtNLAs is an important result, which should be placed into the Fig.1 in the main text, whereas Fig.2 is the subcellular localization of MtNLAs, and so on.
    Our Response: Thank you for your suggestion. We moved the gene structure of NLA homologs and MtNLA paralogs to Figure 1 as suggested and left the sequence alignment in Figure S1 and S2.
  2. Fig.S6: The AM phenotype of MtNLA3-overexpression should be removed into Fig.5, and further recombine this figure.
    Our Response: Thank you for your suggestion. We added the images of colonized root fragments of Ev and overexpressing roots to Figure 6 and 7.

Reviewer 2 Report

Comments and Suggestions for Authors

In the present work Lin and colleagues analyzed the role of NLA genes in Medicago truncatula.

They analyzed 3 homologs (MtLNA1, MtNLA2 and MtNLA3) and three alternative splicing variants of MtNLA3. Even if the work didn’t allow to clearly understand the roles of the genes in response to 

AM symbiosis and nutritional status, it gives some ideas for future works.

The authors did a huge amount of work using different techniques and I found some minor points that must be addressed before publication.

At line 161 the authors wrote they analyzed the expression of MtNRT2;1 without introducing the gene and the reason why they choose to study its expression. It would be very helpful to briefly introduce  the gene before commenting on the results of its expression.

At the same line the authors wrote the expression of MtNRT2;1 was increased under low-nitrate: looking at figure 2B it’s surprising that the expression in the -N condition is statistically identical to the expression in the -P condition but different from the control (+) condition. Can the authors comment on this?

At lines 201-204 the authors used the data on MtBCP1 gene expression to state that there is no difference of colonization efficiency by the two fungal species. The analyzed gene is preferentially expressed in the main trunk of the arbuscules, so in case the arbuscular mycorrhizal (AM) colonization is very mature and the arbuscules are more or less all collapsed and the roots are full of vescicles, the AM colonization will be very high in term of intensity but the expression of MtBCP1 will be low. How can the authors exclude that one of the AM fungi colonized the root faster than the other, and developed a very high number of vescicles in addition to the active arbuscules taken in consideration by the analysis of the expression of MtBCP1, resulting in a higher colonization rate?

In my opinion the molecular data must be supported by a morphological quantification of the AM colonization.

At line 251 the authors presented the experiment on composite plants: how did they select the transformed piece of root from the not transformed? Can the authors add some details on the material and method section?

At line 353 the authors wrote they analyzed the morphology of arbuscules in overexpressing roots: how did they do this? Had the authors some picture at a very high magnification suitable to analyze the fine arbuscule morphology? The images in figure S6 are very poor: the magnification is too low and the pictures seem out of focus. It is difficult to appreciate the fungal colonization and it is almost impossible to see the fine detail of arbuscules. Can the authors provide higher quality images?

Line 371: how were the Medicago seeds sterilized?

Line 429: the authors wrote they followed the Grid method to quantify the AM colonization. As far as I know, the grid method allows to obtain at least 4 parameters: the level of arbuscular colonization, the level of hyphal colonization and the level of vescicular colonization plus a sum of the other three parameters. Which of these parameters was used in the work?

Author Response

Dear reviewer,

We are grateful for the comments and suggestions to improve our manuscript. Here are our point-by-point responses to your comments.

  1. At line 161 the authors wrote they analyzed the expression of MtNRT2;1 without introducing the gene and the reason why they choose to study its expression. It would be very helpful to briefly introduce the gene before commenting on the results of its expression. At the same line the authors wrote the expression of MtNRT2;1 was increased under low-nitrate: looking at figure 2B it’s surprising that the expression in the -N condition is statistically identical to the expression in the -P condition but different from the control (+) condition. Can the authors comment on this?

Our Response: Thank you for your comments. We added the information about MtNRT2;1 as suggested (line 185-187) and corrected the post-hoc labels.

At lines 201-204 the authors used the data on MtBCP1 gene expression to state that there is no difference of colonization efficiency by the two fungal species. The analyzed gene is preferentially expressed in the main trunk of the arbuscules, so in case the arbuscular mycorrhizal (AM) colonization is very mature and the arbuscules are more or less all collapsed and the roots are full of vescicles, the AM colonization will be very high in term of intensity but the expression of MtBCP1 will be low. How can the authors exclude that one of the AM fungi colonized the root faster than the other, and developed a very high number of vescicles in addition to the active arbuscules taken in consideration by the analysis of the expression of MtBCP1, resulting in a higher colonization rate?
Our Response: Thank you for your comments. Actually, we have examined the mycorrhization under microscope and reported in the previous report (Deng et al., 2022). Based on the result of microscopic observation and molecular data, we believed that the colonization efficiency of these two different fungal species was similar.

Reference: Deng, C.; Li, C.J.; Hsieh, C.Y.; Liu, L.Y.D.; Chen, Y.A.; Lin, W.Y. MtNF-YC6 and MtNF-YC11 are involved in regulating the transcriptional program of arbuscular mycorrhizal symbiosis. Front Plant Sci 2022, 13, 976280, doi:10.3389/fpls.2022.976280.

  1. In my opinion the molecular data must be supported by a morphological quantification of the AM colonization.
    Our Response: Thank you for your comments. We added the images of colonized root fragments of Ev and overexpressing roots to Figure 6 and 7 to show the morphology of fungal colonization.
  2. At line 251 the authors presented the experiment on composite plants: how did they select the transformed piece of root from the not transformed? Can the authors add some details on the material and method section?
    Our Response: The construct we used contains a DsRed gene driven by CaMV 35S promoter which can be used as a selection marker. Thus, before transplanting and sample harvesting, we observed the whole root system under fluorescence microscope and trimmed off roots without red fluorescence. The detail was added in the material and method section (line 429-432, 452-453).
  3. At line 353 the authors wrote they analyzed the morphology of arbuscules in overexpressing roots: how did they do this? Had the authors some picture at a very high magnification suitable to analyze the fine arbuscule morphology? The images in figure S6 are very poor: the magnification is too low and the pictures seem out of focus. It is difficult to appreciate the fungal colonization and it is almost impossible to see the fine detail of arbuscules. Can the authors provide higher quality images?
    Our Response: Thank you for your comments. In figure 6 and 7, we added the images of colonized root fragments to give an idea how the transgenes affect AMF colonization. Based on the images, we can see that AMF is able to colonize well in overexpressing roots and the size of arbuscules in overexpressing roots and Ev are similar. Thus, combined the colonization phenotype and molecular data, we conclude that symbiosis were not significantly affected by transgenes.  
  4. Line 371: how were the Medicago seeds sterilized?
    Our Response: Seeds were treated with concentrate sulfuric acid for 5 min, followed by surface-sterilized with 10% commercial bleach for 10 min. Then, seeds were washed by sterilized water to remove extra bleach. The detail was added in the material and method section as suggested (line 406-407).
  5. Line 429: the authors wrote they followed the Grid method to quantify the AM colonization. As far as I know, the grid method allows to obtain at least 4 parameters: the level of arbuscular colonization, the level of hyphal colonization and the level of vescicular colonization plus a sum of the other three parameters. Which of these parameters was used in the work?
    Our Response: Thank you for your comments. Trouvelot et al. (1986) proposed 5 parameters: (1) F% is the frequency of mycorrhization in the root systems (No. of mycorrhizal root fragments/ No. of total root fragments observed); (2) M% is the intensity of mycorrhization in the root systems; (3) m% is the intensity of mycorrhization in mycorrhizal root fragments; (4) A% is the arbuscule abundance in total root system; (5) a% is the arbuscule abundance in mycorrhizal root fragments. To evaluate the intensity of myzorrhization and the arbuscule abundance, people have to score the mycorrhizal root fragments in class 1 to 5. We think that that judgement is subjective. In the past years many AM symbiosis-related papers only use F% as mycorrhization indicator. Thus, we decided to only show F% to reflect AMF colonization efficiency.

Reference: Trouvelot A, Kough JL & Gianinazzi-Pearson V (1986) Mesure du taux de mycorhization VA d’un système radiculaire. Recherche de méthodes d’estimation ayant une signification fonctionnelle. In : Physiological and Genetical Aspects of Mycorrhizae, V. Gianinazzi-Pearson and S. Gianinazzi (eds.). INRA Press, Paris, pp. 217-221.

Reviewer 3 Report

Comments and Suggestions for Authors

In this manuscript (plants-2726494) entitled "Differential responses of Medicago truncatula NLA homologs to nutrient deficiency and arbuscular mycorrhizal symbiosis" submitted to Plants, Wei-Yi Lin and colleagues have investigated the subcellular localization and the responses of MtNLAs to external nutrient status. This research is interesting and convincing, but minor points need to be addressed to improve the quality of this manuscript.

1. For Figure 1, at least three representative cells should be shown for each subcellular localization of MtNLAs in the revised Figure. In addition, the PM marker should be included to indicate the PM in the revision.

2. For Figure 2, At least three different concentrations of Pi and/or nitrate should be employed for the low-Pi or low-nitrate treatments in the revised Figure.

3, For Figure 4, protein accumulation of MtNLAs and MtPT1 or MtPT4 in the yeast cells should be analyzed in the revision.

4, For Figure 5, representative pictures to show the symbiotic phenotypes in MtNLA3.1-overexpressing roots should be displayed in the revised manuscript.

Author Response

Dear Reviewer,

We are grateful for the comments and suggestions to improve our manuscript. Here are our point-by-point responses to your comments.

  1. For Figure 1, at least three representative cells should be shown for each subcellular localization of MtNLAs in the revised Figure. In addition, the PM marker should be included to indicate the PM in the revision.
    Our Response: Thank you for your suggestion. We repeated the experiments at least two times and for each construct we observed more than 10 cells. Unfortunately, we did not take photos for every cell we observed. In Figure S3a, we provided two more images of GFP-MtNLAs. Due to conservation of PM localization of NLA1 homologs in Arabidopsis and rice, thus we did not co-express MtNLAs with a PM marker.
  2. For Figure 2, At least three different concentrations of Pi and/or nitrate should be employed for the low-Pi or low-nitrate treatments in the revised Figure.
    Our Response: Thank you for your suggestion. It has been shown the downregulation of NLA1 homologs to low Pi and low nitrate in rice and Arabidopsis roots (Kant et al. 2011 and Yang et al. 2017). In this study, we aimed to compare the transcriptional responses of MtNLAs to Pi and nitrate treatments with those in rice and Arabidopsis. Thus, we only included 4 nutrient treatments, high P/high N, low P/high N, high P/low N and low P/low N. Surprisingly, MtNLA3 was upregulated by Pi in roots. We plan to study the of regulation of NLA expression in Medicago in the future.

    reference:
    Kant, S.; Peng, M.; Rothstein, S.J. Genetic regulation by NLA and microRNA827 for maintaining nitrate-dependent phosphate homeostasis in Arabidopsis. PLoS Genet 2011, 7, e1002021.
    Yang, J.; Wang, L.; Mao, C.; Lin, H. Characterization of the rice NLA family reveals a key role for OsNLA1 in phosphate homeostasis. Rice 2017, 10, 52.   
  3. For Figure 4, protein accumulation of MtNLAs and MtPT1 or MtPT4 in the yeast cells should be analyzed in the revision.
    Our Response: All the constructs for Y2H were sequenced before transforming to yeast cells. In protein-protein interaction tests, we included positive (NubI) and negative (NubG) controls which indirectly confirm MtPT1 and MtPT4 expression and membrane targeting. The test was repeated at least three times to make sure the consistence of the protein-protein interactions. Thus, we did not check protein accumulation in yeasts.

For Figure 5, representative pictures to show the symbiotic phenotypes in MtNLA3.1-overexpressing roots should be displayed in the revised manuscript.
Our Response: Thank you for your suggestions. We added the images of colonized root fragments from overexpressing and Ev composite roots in Figure 6 and 7.

Reviewer 4 Report

Comments and Suggestions for Authors

This publication presents interesting results related to the effects of Pi and N deficiency and mycorrhizae symbiosis on the differential responses of NLA homologs in Medicago truncatula. The adopted approach in this study was very consistent since it could provide more insights into the different functions orchestrated by NLA genes, especially their involvement in AM symbiosis and plant response to nutrient limitation.

The manuscript was well introduced, and the authors adopted convincing methods with a discussion of the different obtained results. However, the manuscript needs moderate revisions to be suitable for publication in Plants.

General comment

Comment 1: The English of this manuscript needs moderate revision.

Comment 2: Some methods description in M&M section should be more precise.

Comment 3: Some concise conclusions based on your results are missing.

Other comments

- Abstract

L17: “to external nutrient status”, please be more specific.

L20-21: please italicize gene names.

- Introduction

L35-36: Please italicize the transcript name.

L62-68: what about MtPT9 and MtPT10?

L71: “MtPT4 localizes on the periarbuscular membrane”, please correct.

L73: Please provide the significance of “AM” at the first appearance in the text.

- Results

The quality of Figures needs to be improved.

L145: Please provide the significance of GFP in the Figure caption.

L190: Please correct the title of Fig. 2 by adding Pi concentration.

L192-193: “Different letters indicate significant difference (p<0.05)” please add according to which test. The same comment of the other figures caption.

L259: Why did you assess the frequency of mycorrhization in Fig. 5a?? It would be better to assess its intensity per example.

- Discussion

Please provide some explanation to the difference recorded in the response of MtNLA3 variants in R. irregularis- and C. etunicatum-colonized roots.

- M&M

L369: please provide more details about the growth conditions.

L373-376: Please specify all the applied treatments, it is not clear.

L378: Based on what did you apply Claroideoglomus etunicatum or Rhizophagus irregularis strains?

L388: Please change “Table 1 Suppl” to “Table S1”. Please check throughout the manuscript.

L406: Please provide the significance of “MES” at the first appearance.

L429: Please change “as described [40]” to “as previously described [40]”.

- Conclusions

Please provide concise conclusions, at the end of the Discussion section, based on the obtained results

Comments on the Quality of English Language

The English of this Manuscript needs moderate edition.

Author Response

Dear Reviewer,

We are grateful for the comments and suggestions to improve our manuscript. Here are our point-by-point responses to your comments.

  1. The English of this manuscript needs moderate revision.
    Our Response: Thank you for your suggestion. We proofread the manuscript. The manuscript was edited by English editing service.
  2. Some methods description in M&M section should be more precise.
    Our Response: Thank you for your suggestion. We made the content more precise.
  3. Some concise conclusions based on your results are missing.
    Our Response: We summarized our findings in the conclusion section.
  4. L17: “to external nutrient status”, please be more specific.
    Our Response: We modified the sentence as “to external Pi and nitrate status.”
  5. L20-21: please italicize gene names.
    Our Response: The gene name is italicized.
  6. L35-36: Please italicize the transcript name.

Our Response: The gene name was italicized.

  1. L62-68: what about MtPT9and MtPT10?
    Our Response: After checking MtPTs in NCBI database, we found the misleading information provided by Cao et al. (2021) which mentioned 10 PTs in Medicago. In fact, there was only 9 members in Medicago. Cao et al. (2021) showed that the transcript levels of MtPT9 was very low although it was upregulated by low Pi. We corrected the information about MtPTs in the introduction. 

Reference: Cao, Y.M.; Liu, J.L.; Li, Y.Y.; Zhang, J.; Li, S.X.; An, Y.R.; Hu, T.M.; Yang, P.Z. Functional analysis of the phosphate transporter gene MtPT6 from Medicago truncatula. Front Plant Sci 2021, 11, 620377.

  1. L71: “MtPT4localizes on the periarbuscular membrane”, please correct.
    Our Response: MtPT4 has been shown to localize on the periarbuscular membrane in Harrison et al. (2002). Periarbuscular membrane indicates a specialized plasma membrane of the inner cortical cell that encircles an arbuscule, a specialized fungal structure.

Reference: Harrison, M.J.; Dewbre, G.R.; Liu, J. A phosphate transporter from Medicago truncatula involved in the acquisition of phosphate released by arbuscular mycorrhizal fungi. Plant Cell 2002, 14, 2413-2429.

  1. L73: Please provide the significance of “AM” at the first appearance in the text.
    Our Response: We added the few sentences to show the importance of AMF on plants (line 59-61).

 - Results

  1. The quality of Figures needs to be improved.
    Our Response: The quality of figures is improved.
  2. L145: Please provide the significance of GFP in the Figure caption.
    Our Response: The information is provided in the figure caption.
  3. L190: Please correct the title of Fig. 2 by adding Pi concentration.
    Our Response: The information is provided in the figure caption.
  4. L192-193: “Different letters indicate significant difference (p<0.05)” please add according to which test. The same comment of the other figures caption.
    Our Response: The data was evaluated by ANOVA with Tukey’s HSD test. The test we used is provided in the figure caption.
  5. L259: Why did you assess the frequency of mycorrhization in Fig. 5a?? It would be better to assess its intensity per example.
    Our Response: Trouvelot et al. (1986) proposed another 4 parameters to evaluate AM symbiosis which require to score the phenotype by personal judgement. Now, many papers including this manuscript prefer to evaluate the frequency of mycorrhization to see whether fungi can colonize host plant successfully. We think it is objective measurement, compared to another 4 parameters. Combined with phenotype of mycorrhizal roots and molecular analysis of symbiosis marker genes, the readers can understand the effects of transgenes on symbiosis.

Reference: Trouvelot A, Kough JL & Gianinazzi-Pearson V (1986) Mesure du taux de mycorhization VA d’un système radiculaire. Recherche de méthodes d’estimation ayant une signification fonctionnelle. In : Physiological and Genetical Aspects of Mycorrhizae, V. Gianinazzi-Pearson and S. Gianinazzi (eds.). INRA Press, Paris, pp. 217-221.

- Discussion

Please provide some explanation to the difference recorded in the response of MtNLA3 variants in R. irregularis- and C. etunicatum-colonized roots.
Our Response: In this paragraph we have mentioned that the differential expression of symbiotic-responsive genes when colonized by different AMF species has been observed in Medicago, cassava and sorghum. These findings indicate that the great impacts of genetic background of fungal species on the plant-AMF interaction.

Reference:
Grunwald, U.; Guo, W.; Fischer, K.; Isayenkov, S.; Ludwig-Muller, J.; Hause, B.; Yan, X.; Kuster, H.; Franken, P. Overlapping expression patterns and differential transcript levels of phosphate transporter genes in arbuscular mycorrhizal, Pi-fertilised and phytohormone-treated Medicago truncatula roots. Planta 2009, 229, 1023-1034.

Mateus, I.D.; Masclaux, F.G.; Aletti, C.; Rojas, E.C.; Savary, R.; Dupuis, C.; Sanders, I.R. Dual RNA-seq reveals large-scale non-conserved genotype x genotype-specific genetic reprograming and molecular crosstalk in the mycorrhizal symbiosis. ISME J 2019, 13, 1226-1238.
Watts-Williams, S.J.; Emmett, B.D.; Levesque-Tremblay, V.; MacLean, A.M.; Sun, X.; Satterlee, J.W.; Fei, Z.; Harrison, M.J. Diverse Sorghum bicolor accessions show marked variation in growth and transcriptional responses to arbuscular mycorrhizal fungi. Plant Cell Environ 2019, 42, 1758-1774.

- M&M

  1. L369: please provide more details about the growth conditions.
    Our Response: The information is provided as suggested (line 404-405).
  2. L373-376: Please specify all the applied treatments, it is not clear.
    Our Response: The information is provided as suggested (line 409-411).
  3. L378: Based on what did you apply Claroideoglomus etunicatumor Rhizophagus irregularis strains?
    Our Response: Rhizophagus irregularis is widely used for analyzing symbiotic relationships with many plant species, including Medicago. Claroideoglomus etunicatum is another AMF species that can colonize with a wide range of plant species and enhance plant stress tolerance.
  4. L388: Please change “Table 1 Suppl” to “Table S1”. Please check throughout the manuscript.
    Our Response: We have corrected and checked throughout the text.
  5. L406: Please provide the significance of “MES” at the first appearance.
    Our Response: MES is used for buffering the solution. The information is provided.
  6. L429: Please change “as described [40]” to “as previously described [40]”.
    Our Response: We have modified the sentence as suggested.

- Conclusions

Please provide concise conclusions, at the end of the Discussion section, based on the obtained results
Our Response: Thank you for your suggestions. We concluded the results in the end of Discussion section.

Round 2

Reviewer 2 Report

Comments and Suggestions for Authors

In the revised version of the article the authors partially solved the criticism I raised. Two major points remained unsolved and must be addressed before pubblication.

  1. The quality of the images of colonized roots in figures 6 and 7 is very low. The magnification is low and didn’t allow to appreciate the fine morphology of the arbuscules. The only thing the readers can appreciate is the arbuscules distribution but not the morphology. If the purpose of the images is to show the morphology of a single arbuscules, the authors must insert images of single arbuscule at higher magnification and with a high level of definition

  1. The authors wrote in the article they evaluated the level of AM colonization following the grid method (reference given in the article: Mcgonigle, T.P.; Miller, M.H.; Evans, D.G.; Fairchild, G.L.; Swan, J.A. A new method that gives an objective measure of colonization of roots by vesicular-arbuscular mycorrhizal fungi. New Phytol 1990, 115, 495-501, doi:10.1111/j.1469- 8137.1990.tb00476.x.) while in the response to the reviewer they wrote they used the Trouvelot method and gave the reference of Toruvelot et al., 1986. I’m confused: the two methods are deeply different, which method did they use?? The method written in the article or the other written in the letter in response to the referees comment?

Author Response

Dear Reviewer,

We are grateful for the comments to improve our manuscript. Here are our point-by-point responses to your comments. Hope you will be satisfied with our resubmission.

  1. The quality of the images of colonized roots in figures 6 and 7 is very low. The magnification is low and didn’t allow to appreciate the fine morphology of the arbuscules. The only thing the readers can appreciate is the arbuscules distribution but not the morphology. If the purpose of the images is to show the morphology of a single arbuscules, the authors must insert images of single arbuscule at higher magnification and with a high level of definition
    Our response: Thank you for your comments. The purpose of these images is to show that the distribution of arbuscules in colonized roots in Ev and overexpressing roots is similar. We are sorry that the root samples were left in PBS for 2-4 years and are not suitable for sectioning for observing the morphology of arbuscules. Thus, we modified the text and mentioned that there is no effect of transgenes on the distribution instead of arbsucle development. Hope you can understand the difficulties and will be satisfied with our revision.
  2. The authors wrote in the article they evaluated the level of AM colonization following the grid method (reference given in the article: Mcgonigle, T.P.; Miller, M.H.; Evans, D.G.; Fairchild, G.L.; Swan, J.A. A new method that gives an objective measure of colonization of roots by vesicular-arbuscular mycorrhizal fungi. New Phytol 1990, 115, 495-501, doi:10.1111/j.1469- 8137.1990.tb00476.x.) while in the response to the reviewer they wrote they used the Trouvelot method and gave the reference of Toruvelot et al., 1986. I’m confused: the two methods are deeply different, which method did they use?? The method written in the article or the other written in the letter in response to the referees comment?
    Our response: We are sorry about the confusion. We evaluated the mycorrhization by grid method (Mcgonicgle et al., 1990) in this manuscript. Your previous question mentioned about at least four parameters allowed to be evaluated and these parameters were defined by Trouvelot et al. (1986). When observing the root fragments under stereomicroscope, we only count the percentage of colonized root fragments and the percentage of colonized root fragments containing arbuscules. We think the evaluation is subjective, compared to other parameters which rely on human rating. Many publications use the same way to show AMF colonization. Thus, we did not further score the colonization and the abundance of arbuscules.

Reviewer 4 Report

Comments and Suggestions for Authors

The authors satisfied all the raised comments; thereby, I recommend the publication of the current version of the manuscript.

Author Response

Dear reviewer,

Thank you for consideration for publication in Plants.

Sincerely yours,

Wei-Yi Lin